# What can 5.17 billion regression fits tell us about artificial models of the human visual system?

**Colin Conwell,**[*] **Jacob S. Prince, George A. Alvarez, Talia Konkle**
Department of Psychology, Harvard University

## Abstract

Rapid simultaneous advances in machine vision and cognitive neuroimaging present an unparalleled opportunity to (re)assess the current state of artificial models of the human visual system. Here, we perform a large-scale benchmarking analysis of 85 modern deep neural network models (e.g. CLIP, BarlowTwins, Mask-RCNN) to characterize with robust statistical power how differences in architecture and training task contribute to the prediction of human fMRI activity across 16 distinct regions of the human visual system. We find: one, that even stark architectural differences (e.g. the absence of convolution in transformers and MLP-mixers) have very little consequence in emergent fits to brain data; two, that differences in task have clear effects–with categorization and self-supervised models showing relatively stronger brain predictivity across the board; three, that feature reweighting leads to substantial improvements in brain predictivity, without overfitting – yielding model-to-brain regression weights that generalize at the same level of predictivity to brain responses over 1000s of new images. Broadly, this work presents a lay-of-the-land for the emergent correspondences between the feature spaces of modern deep neural network models and the representational structure inherent to the human visual system.

## 1 Introduction

The sheer pace of progress in computer vision poses a practical challenge for neuroscientists assessing state-of-the-art models in their ability to explain visual representation and behavior. New high-performing models are released on a near-daily basis, and recent innovations (e.g. in self-supervised learning [Grill et al., 2020; Zbontar et al., 2021; He et al., 2020; Caron et al., 2020, 2021]) have created myriad new opportunities for productive synergy between the fields of biological and machine vision. As such, methods for comparing the brain-predictivity of artificial models using predefined pipelines ("neural benchmarking") could prove critical in helping discern the algorithmic innovations that may be meaningful with respect to the study of brain function (Yamins et al., 2014; Khaligh-Razavi & Kriegeskorte, 2014; Cadieu et al., 2014; Kriegeskorte, 2015; Güçlü & van Gerven, 2015; Cichy et al., 2016; Eickenberg et al., 2017; Schrimpf et al., 2018; Wen et al., 2018; Cadena et al., 2019; Bashivan et al., 2019; Serre, 2019; Cadena et al., 2021; Marques et al., 2021; Storrs et al., 2021).

Existing public neural benchmarking efforts have been limited to mouse and primate neurophysiology (Schrimpf et al., 2018; Conwell et al., 2021). Recent advances in the scale and quality of human neuroimaging datasets (Chang et al., 2019; Allen et al., 2021; Contier et al., 2021) provide testbeds for assessing the state of deep neural network modeling as applied to the human visual system.

Here we present a large-scale benchmark of dozens of state-of-the-art deep neural network models in their prediction of human brain activity across the visual hierarchy. Aiming for coverage, our survey

---

[*]Correspondence: conwell@g.harvard.edu; Project Website: github.com/ColinConwell/DeepNSD

3rd Workshop on Shared Visual Representations in Human and Machine Intelligence (SVRHM 2021) of the Neural Information Processing Systems (NeurIPS) Conference, Virtual.

attempts to document the current trends in how well different kinds of models, varying in both task and architecture, learn features with brain-like response signatures. Our results complement prior work examining different model predictivities and different methods of neural mapping (Khaligh-Razavi & Kriegeskorte, 2014; Khaligh-Razavi et al., 2017; Wang et al., 2019; Storrs et al., 2021), but at a significantly larger scale, and incorporating a set of more modern models not yet fully accounted for in the benchmarking literature (e.g. self-supervised models, and vision transformers).

## 2    Methods

As the target of our neural benchmark, we use the Natural Scenes Dataset (NSD, [Allen et al., 2021]), a recent fMRI dataset representing the most extensive sampling of visual responses in individual participants to date (30,000 stimuli viewed per subject; 73,000 unique images total). Here we analyze only a small fraction of this dataset, focusing on responses to 1,000 COCO (Lin et al., 2014) stimuli that were shown to 4 subjects at least 3 times, in a subset of ROIs along the visual hierarchy. We compare these responses with the responses of 83 modern DNNs that vary in task and architecture.

We employ two methods for mapping the activations of model features within a layer to regions of the brain – classical representational similarity analysis (RSA, [Kriegeskorte et al., 2008a]) and voxelwise-encoding (re-weighted) RSA (Konkle & Alvarez, 2021; Kaniuth & Hebart, 2021). Classical RSA considers all of the features from a given model layer equally in computing the image-wise representational dissimilarity matrix (RDM), which is directly compared with a target brain RDM. This method requires a fully-emergent match in population-level geometry between a neural ROI and the full set of units in a model layer.

Voxelwise encoding RSA (veRSA), on the other hand, takes advantage of feature reweighting to identify different model subspaces that correspond to the variance in different brain regions. To implement voxel-wise encoding RSA, we use an efficient high-throughput model-fitting procedure, first applying leave-one-out cross-validated ridge regression to map between a given model feature space and the observed univariate activity pattern of each voxel; once we've collected a set of predictions for the patterns of activity for each voxel in a given ROI, we compute an RDM from these predictions and compare that RDM to the RDM in the brain. This re-weighted RSA procedure requires massive parallelization, and entails performing a total of around 5.17 billion regression fits (calculated by multiplying the total number of model layers we analyze by the total number of voxels under consideration from the brain dataset). To assuage concerns of overfitting, we validate the robustness of our fitted regressions by testing their generalizability to 1000 independent images (specific to each subject) entirely removed from the training procedure.

More detailed descriptions of the brain data A.1, candidate model architectures A.2, and mapping procedures A.3 may be found in the Appendix.

## 3    Results

### 3.1    Hierarchical Correspondence

As a first step and sanity check, we ask: Does the seminal finding that the information processing hierarchies in deep nets recapitulate the information processing hierarchy in the human visual system (Yamins et al., 2014; Güçlü & van Gerven, 2015, 2017) hold at scale and across a significantly diverse population of models? The answer is a strong affirmative (**Figure 1**): Using a purely data-driven aggregation procedure, we show that the relative depth of the best-fitting model layer for each ROI seems to re-capitulate the human visual hierarchy (e.g. early visual areas, followed by category-selective regions). This hierarchical convergence holds even when breaking down the models by broad, divergent classes of architecture.

### 3.2    Architecture Variation

How do models with different architectures compare in their ability to predict the structure of human brain responses across the visual system? Our particular survey of models, chosen deliberately to reflect the diversity of modern object recognition (ImageNet-trained) architectures, allows for numerous subdivisions, but perhaps the most prominent is between convolutional architectures (e.g.

Figure 1: *(A) Visualization of selected regions-of-interest on a flattened hemisphere. (B) Emergent hierarchical correspondence between the most predictive model layer and the hypothesized information processing hierarchy of the visual system. Regions along the x-axis are ordered by the average depth of the best predicting layer (across all models). Data are also broken down by the architectural distinctions of ConvNets, MLP-Mixers, and Transformers. Each point is the best performing layer from a given model, averaged over subjects.*

VGG, ResNet, MobileNet, $n = 24$), vision transformers (e.g. Visformer, DeIT, $n = 13$) and MLP-mixers (e.g. ResMLP, gMixer, $n = 5$). The latter two of these are more recent advents of computer vision, and are defined by the lack of a convolutional inductive bias – once considered a cornerstone of the link between biological and machine vision. Comparisons between these architectures (across both classical RSA and voxel-encoding RSA) are shown in Figure 2.

To test for differences in predictivity, we use nonparametric ANOVAs. Without reweighting (classical RSA), there is a significant difference across ConvNets, MLP-Mixers, and Transformers $(\chi^2_{\text{Kruskal-Walils}}(2) = 10.14, p_{\text{Holm}} = 0.02, \hat{\varepsilon}^2_{\text{ordinal}} = 0.27, CI_{95\%}[0.09, 0.49])$ in early visual areas, driven by a significant pairwise advantage of ConvNets over Transformers. With reweighting (veRSA), this difference disappears. Without reweighting, there is no significant difference between architectures in higher-level cortical areas. With reweighting, there is a difference $(\chi^2_{\text{Kruskal-Wallis}}(2) = 10.59, p_{\text{Holm}} = 0.02, \hat{\varepsilon}^2_{\text{ordinal}} = 0.26, CI_{95\%}[0.09, 0.56])$, driven this time by the pairwise superiority of both ConvNets and MLP-Mixers over Transformers.

Behind these apparently significant effects is the numerical reality that the raw effect sizes in both cases is effectively negligible – less than $r_{\text{Pearson}} = 0.01$ and $r_{\text{Pearson}} = 0.02$, respectively. As such, the most striking effect here is not that of architecture, but of mapping method, which substantially augments the predictive power of every model in our survey (with average gains of $r_{\text{Pearson}} = 0.160, CI_{95\%}[0.152, 0.166]$ across model and ROI). In the most notable case, models in EBA experience average gains of $r_{\text{Pearson}} = 0.265$. These improvements dwarf any difference attributable to architecture, and underscore an important point: despite dramatic differences in the design and algorithmic inductive biases of ConvNets, MLP-Mixers, and Transformers, there is little consequence on the resulting brain predictivity (regardless of mapping method).

### 3.3 Task Variation

How does brain predictivity vary as a function of task? To answer this empirically, we consider the 24 models from the Taskonomy project (Zamir et al., 2018). These models share the same base architecture (ResNet-50) and training data (visual diet), but are trained on 1 of 24 popular computer vision tasks. These tasks are organized into 4 different categories (2D, 3D, Semantic and Geometric) according to what the authors of the Taskonomy project call the models' 'transfer affinity' – the degree to which a model trained on one task supports transfer learning to another. The predictivity scores of these models for both classical and voxel-wise encoding RSA are shown in Figure 3.

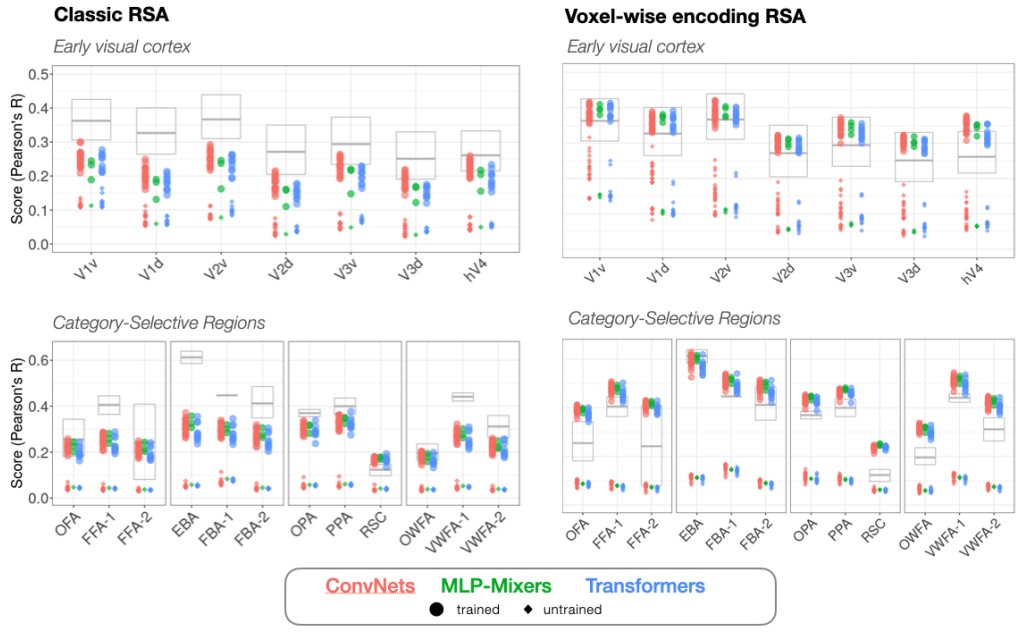

Figure 2: *Architecture variations. Model fits are shown along the y-axis, for early visual areas (top row) and category-selective areas (bottom row), for classical RSA (left) and voxel-wise encoding RSA (right). The gray boxes indicate an intersubject reference point (the average pairwise correlation of individual subject RDMs (for details, see A.5). Each dot is the best performing layer from a single model, with trained models in large circular points, and untrained counterparts in small diamonds.*

Without reweighting, there is considerable variability across ROI in the tasks that are most predictive of the brain, but the differences between the best task and the second-best task is minimal in most cases. In V2, for example, a 2D task (edge detection) is the most predictive of the tasks at $r_{\text{Pearson}} = 0.226$, but is closely followed by a 3D task (Keypoints) at $r_{\text{Pearson}} = 0.220$.

As in the case of architecture, feature reweighting (veRSA) leads to uniform improvement across models. Strikingly, however, object and scene classification gain disproportionately. The gains for object recognition are so substantial that it becomes the single most predictive task for all brain areas, often dominating by an impressively large margin, with a mean gain over the next best task (apart from scene classification) of $r_{\text{Pearson}} = 0.127, CI_{95\%}[0.122, 0.131])$ across all ROIs.

While these results point strongly to an advantage of category supervision in the formation of neurally predictive representation (at least in the case of veRSA), the self-supervised models (absent from Taskonomy) in our survey allow us to delve more deeply into whether the classification objective is the key driver of neural predictivity, or whether category-supervised models derive their advantage from the set of invariances that they learn in service of classification.

The predictive power of our self-supervised models strongly suggest the latter: regardless of mapping method, self-supervised models (especially recent contrastive ResNet-50 models such as SimCLR and BarlowTwins) tend to show a small but statistically significant advantage over a (recently revamped) category-supervised ResNet-50 (Wightman et al., 2021). For example, averaging across brain ROIs, SimCLR eeks out a mean gain of $r_{\text{Pearson}} = 0.013, CI_{95\%}[0.0106, 0.0192]$ in weighted RSA and a gain of $r_{\text{Pearson}} = 0.006, CI_{95\%}[0.002, 0.008]$ in classical RSA. The CLIP models (ResNet and ViT variants trained via contrastive learning on millions of image-text pairs scraped from the internet) were consistently, although by a narrow margin (mean $r_{\text{Pearson}} = 0.0048, CI_{95\%}[0.001, 0.009]$), the best performing model with weighted RSA in category-selective visual areas – achieving the best score of any model tested in 10 / 12 ROIs. While these results should not be interpreted as indicating *superiority* of self-supervision over category-supervision, they do begin to indicate *parity* in prediction levels – a win for ethological plausibility (Konkle & Alvarez, 2021; Zhuang et al., 2021).

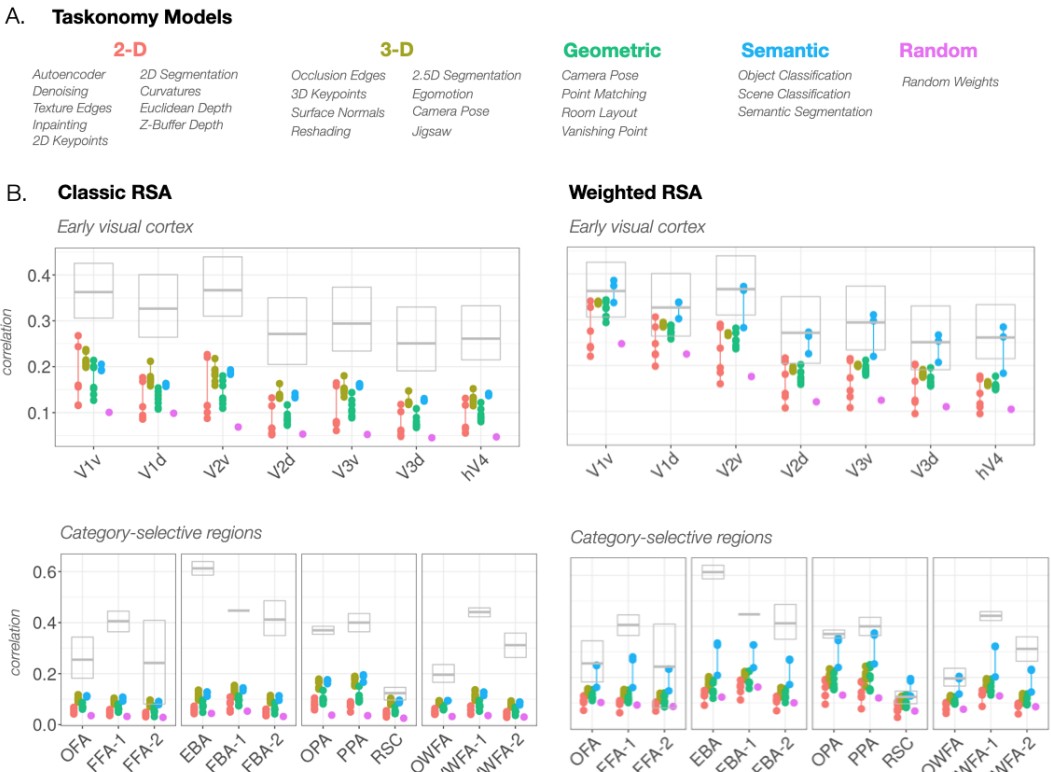

Figure 3: *Effect of task on model-brain predictivity. (A) 24 Taskonomy models with a ResNet50 architecture, grouped into 4 categories (2D, 3D, Geometric, or Semantic) and an untrained (randomly-initialized) model, for comparison. The correlation between the model features and brain responses for early visual areas (top) and category-selective regions (bottom) is plotted, using classical RSA in (B) and reweighted RSA in (C). Gray box plots indicate the range of inter-subject RDM correlations using classical RSA (for further details on this 'reference point' estimate, see A.5).*

## 3.4 Input Variation

While the models in our survey do not allow for a fully-controlled comparison of brain predictivity across varying kinds of input or visual diet – at least, not while also holding architecture and task constant – there is some evidence of its effect implicit in a number of our findings. Foremost among these is the raw numerical difference in predictivity scores between models trained on the ImageNet dataset (i.e. all models discussed in Section 3.2) and the Taskonomy models (3.3). Taskonomy models are trained on a custom image set that consists of a camera moving smoothly through a number of indoor scenes (Zamir et al., 2018). (Taskonomy describes this dataset as consisting of 4.5 million scenes across 600 buildings, at a resolution of 1024x1024 per scene). The average score of ImageNet-trained models across all ROIs is $r_{\text{Pearson}} = 0.240, CI_{95\%}[0.234, 0.246]$ for classic RSA and $r_{\text{Pearson}} = 0.411, CI_{95\%}[0.406, 0.415]$ for weighted RSA. The average score of Taskonomy models is substantially lower, at $r_{\text{Pearson}} = 0.106, CI_{95\%}[0.094, 0.118]$ for classical RSA and $r_{\text{Pearson}} = 0.181, CI_{95\%}[0.163, 0.199]$ for weighted RSA. Scrutinizing object classification alone, ResNet50 trained on ImageNet attains mean scores of $r_{\text{Pearson}} = 0.244$ and $r_{\text{Pearson}} = 0.415$ on classic and weighted RSA, respectively; ResNet50 trained on the Taskonomy dataset attains $r_{\text{Pearson}} = 0.139$ and $r_{\text{Pearson}} = 0.301$. This means that despite being trained on approximately 3x the quantity of images in ImageNet, the models of Taskonomy suffer significantly from the lack of variation[2] in training as it pertains to downstream brain predictivity. Overall, these observations leave open a clear target for a better understanding of which factors in a dataset are relevant (size, diversity, naturalism, number of categories), and why.

---

[2]Analysis by the Taskonomy authors suggests that only 100 of the 1000 imagenet classes are present across the 'scenes' of the Taskonomy dataset – suggesting the difference in object recognition scores may be at least partially attributable to impoverished category variation.

## 3.5 Additional Generalization Tests

The sheer quantity of regression fits required to summarize the predictive performance of our model set, and the vast number of dimensions relative to data points, may raise concern: is this deep encoding pipeline massively overfitting, in spite of our generalized cross-validation procedures? Or, are the estimates we derive truly a reasonable approximation (given the linking assumptions inherent in the analysis) of a given model's brain predictivity?

To address this concern, we conducted a separate generalization test for each model, in which we selected the best performing layer (according to the original LOOCV score from our regression procedure) per subject, per ROI. For these layers, we then use reweighted RSA to compare brain and model feature spaces using a set of 1000 entirely held-out test images per subject. These images were never referenced or incorporated during training, and prediction scores on these images thus provide a measure of "pure" generalization.

Even with this more stringent test, we found little-to-no drop in accuracy in predicting brain representation evoked by the 1000 unique test images per fMRI subject. When aggregating across subjects, models, and ROIs, for example, the mean decrease in score on the unseen images was less than 1% ($r_{\text{Pearson}} = 0.0095, CI_{95\%}[0.00422, 0.0153]$). By adding a mere 103 million regression fits to our initial total of 5.17 billion, then, we can thus confirm definitively that our encoding models generalize to previously-unseen data. (A more detailed figure showing generalization across specific subjects and ROIs is shown in Section A.6 of the Appendix.)

## 4 Discussion

So what can we learn about the human visual system from 5.17 billion regression fits (not to mention many thousands of correlations)? Broadly, it seems, there are two sets of answers, one more pessimistic, one more optimistic. On the side of pessimism, the lack of variation across architecture suggests that groundbreaking innovations in computer vision may often yield little to no change in our ability to predict the representational structure of biological vision, disrupting what was once prophesied to become a glorious feedback loop between neuroscientific insight and engineering innovation. What's more, the frequent variability in interpretation across mapping method seems a potential pitfall if not accounted for with greater vigilance and attention to theoretical commitment. On the side of optimism, it does appear that more general, algorithmic correspondences between DNNs and brains (especially in terms of the information processing hierarchy) persist in spite of an increasingly rapid shift *away from biological plausibility* in computer vision. In opposite direction of this shift is a promising move *towards ethological plausibility*: Many cutting-edge models no longer rely on learning targets humans almost certainly do not share (e.g. full category supervision). Not coincidentally, these models appear to be competitive predictors of brain activity.

Abstracting away for a moment from specific findings, one overarching question that we hope our analysis could address is: what is the real value and ideal format of neural benchmarking analyses? Our thesis in some sense is that pure benchmarking analyses are ideally supplemented by in-depth analyses of the critical design choices a modeler might make if their end goal is predicting the brain. To this end, we designed this benchmarking analysis specifically with certain kinds of contrasts in mind – namely, architecture and task variation (and to some extent, albeit implicitly, input variation). Our goal in this benchmarking effort was not, therefore, to 'discover' the single best model of the human visual system, but to open the door for further inspection of what kinds of models do well in different parts of the visual system and why. It seems unlikely that models optimized for computer visual tasks alone will cover all the biological visual system's idiosyncrasies; but it seems equally unlikely that models optimized to predict biological vision will remain in sync with the kinds of engineering advances that could approximate critical biological processes in silico. The goal of our benchmarking, then, is to arbitrate what kinds of engineering advances seem to matter and which don't – something we can only (pending theoretical advances) determine empirically.

Current models are still far from capturing the kaleidoscopic complexity of biological visual systems assayed at scale. To build next-generation perceptual models will almost certainly require a tighter linking of neuroscience and engineering, with benchmarks that assess not only raw neural predictivity, but behavioral (psychophysical) and organizational (topographic) similarity, all for a fuller picture of what makes the biology so capable in so many divergent domains.

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

## A  Appendix

### A.1  Human Brain Data

The Natural Scenes Dataset (NSD) is the largest effort to date to measure human brain responses with functional magneic resonance imaging (fMRI), reflecting measurements of 73,000 unique stimuli from the Microsoft Common Objects in Context (COCO) dataset ((Lin et al., 2014)) at high resolution (7T field strength, $1.33s$ TR, $1.8mm^3$ voxel size). In the present work, we analyze only a small fraction of this dataset, focusing on responses to images that enable direct comparison between data from different subjects. That is, we focus on the 1000 COCO stimuli that overlapped between subjects (the "shared1000" images), and limit analyses to the 4 subjects (subjects 01, 02, 05, 07) for whom all 3 image repetitions are available for the shared1000. For the generalization tests, we also select a random unique set of 1000 images for each subject; these were not included in the "shared1000." All responses were estimated using a custom GLM toolbox ("GLMsingle" (Prince et al., 2021)), which was applied during the preprocessing of NSD time-series data, featuring optimized denoising and regularization procedures, to accurately measure changes in brain activity in response to each experimental stimulus.

We focus our analyses to voxels within a set of predefined functional ROIs that span the visual hierarchy (see (Allen et al., 2021) for details on the procedures used to define the ROIs). Further, to maximize SNR of the target data, we implement a reliability-based voxel selection procedure (Tarhan & Konkle, 2020) that isolates regions of the brain containing stable structure in their responses. To compute the split-half reliability a given voxel, we use 1,000 images from each subject (independent from the shared1000, and from all images included in our main analyses and generalization tests), and take the average correlation in univariate response profiles over each pair of available image repetitions (e.g. $mean(r(rep1, rep2), r(rep2, rep3), r(rep1, rep3))$). ROI voxels

exceeding a reliability threshold of $r_{\text{Pearson}} = 0.1$ were included in subsequent analyses. These procedures yield a matrix of dimension *images x voxels x repetitions* for each subject's ROI, and we average over the 3 repetitions to yield the final ROI data input into our neural benchmarking pipeline.

## A.2 Candidate Deep Neural Network Models

In total, we survey a set of 85 distinct models (123 total, when also including the randomly-initialized models). These models are sourced from six different repositories: the Torchvision (PyTorch) model zoo (Paszke et al., 2019); the pytorch-image-models (timm) library (Wightman, 2019); the VISSL (self-supervised) model zoo (Goyal et al., 2021); the CLIP collection (Radford et al., 2021); the Taskonomy (visualpriors) project (Zamir et al., 2018; Sax et al., 2018, 2019); and the Detectron2 model zoo (Wu et al., 2019). The first two of these repositories offer pretrained versions of a large number of object recognition models with varying architectures: including (classic and modern) convolutional networks, vision transformers, and MLP-mixers. For each of these 'ImageNet' (object recognition) models, we include one trained and one randomly initialized variant (using whatever initialization scheme the model authors recommend) so as to assess the impact of ImageNet training on brain prediction, and as a sanity check. The self-supervised models are mainly variants on a popular convolutional architecture (ResNet-50), though do include some transformers (the DINO and CLIP ViT models). The Taskonomy models consist of a core encoder-decoder architecture trained on 24 different common computer vision tasks, ranging from autoencoding to edge detection. These models are engineered in such a way that only the architecture of the decoder varies across task, allowing us to assess (after detaching the encoder) what effect different kinds of training has on brain predictivity, independent of model design.

## A.3 Benchmarking Pipeline

**Feature Extraction** For each of our deep neural network models, we extract features in response to each of our probe stimuli at each distinct layer of the network. Importantly, we define a layer here as a distinct computational (sub)operation. This means, for example, that we treat convolution and the rectified nonlinearity that follows it as two distinct feature maps. This is especially relevant in the case of transformers, where the features inherent to the key - query - value computation of the attention maps often differ substantially. At the end of our feature extraction procedure, we have (for each model and each model layer) a matrix of features of the dimensions *number of images x number of flattened features from the target layer*.

**Classical RSA (cRSA)** As a first method of mapping deep neural network responses to voxel responses, we use classical RSA, a nonparametric mapping method that quantifies the *emergent* similarity of the 'representational geometry' between two feature spaces, regardless of origin. To compute this metric, we construct representational dissimilarity matrices (RDMs) using the pairwise correlation distance (1 - Pearson's $r$) between the responses of a given brain ROI (image by voxel) or a given model layer (image by unit) for all images being considered. We then compare these RDMs by taking a second-order correlation (Pearson's $r$) between the flattened upper-triangular portion of each. This ultimately yields a matrix of correlation scores of dimensions *number of subjects x number of ROIs x number of model layers x number of models*. Classical RSA reflects the extent to which the representational structure in each model layer naturally recapitulates the representational structure in a visual cortical ROI, without alteration or feature reweighting.

**Voxelwise Encoding RSA** The following procedure yields the billions of regression fits we reference in the title. The pipeline works as follows: first, we fit a regression for *each voxel* as a weighted combination of model layer features. Given that the number of features in a layer sometimes number in the millions, we employ sparse random projection (Li et al., 2006) as a dimensionality-reduction procedure, and then use ridge regression as a linear model to relate the model feature space to each voxel's tuning function. Then, we use the voxel-encoding models to generate predicted activation profiles to the complete set of held-out images, and correlate the subsequent predicted representational similarity structure to that of the brain. For additional detail, see Section A.4.

We emphasize that this method contrasts with popular practices in primate and mouse benchmarking, which treat predictivity of unit-level univariate response profiles as the key measure. However, fMRI affords more systematic spatial sampling over the cortex. Thus, for the present analysis, rather than taking the aggregate of single voxel fits as our key measure, we choose to treat the population

representational geometry over each ROI as our critical target for prediction. This multi-voxel similarity structure provides different kinds of information about the format of population-level coding than do individual units (Kriegeskorte et al., 2008b).

**Noise Ceilings and Reference Metrics**

While powerful in the quantity and diversity of its images, the number of repetitions in image presentation (3 per image) in the NSD dataset leaves little room to estimate a noise ceiling per voxel with standard split-half reliability methods. Thus, as a reference metric for how well our models are doing overall, we use inter-subject predictivity: a measure of how well the brain of one human predicts the brain of another. Here, we took the average pairwise correlation of the individual subject RDMs in a given ROI. For a more in-depth discussion of ceilings and reference points, as well as experimental alternatives, see Section A.5.

## A.4 Voxelwise Encoding RSA In-Depth

To predict the activity profile of each voxel, we first use Sparse Random Projection (SRP) (Li et al., 2006) to project the model features generated in response to our 1000 probe images into a lower-dimensional space. We use a dimensionality of 5960 projections–a number we chose *a priori* using the Johnson-Lindenstrauss lemma, which mathematically guarantees the preservation of pairwise distances in a given space of operations (with a minimal distortion defined by a hyperparameter epsilon, which we leave in all cases at the scikit-learn default of 0.1). We then perform a leave-one-out cross-validated (LOOCV) ridge regression (cross-validating over images) to map these projections to the responses of our voxels, obtaining a vector of predicted voxel responses that we then correlate with the true voxel responses to obtain a score per voxel per model layer[3].

This leave-one-out cross-validation is performed in a single matrix operation often referred to as generalized cross-validation, and is numerically equivalent to iterative leave-one-out, but is effectively instantaneous. We iterate this regression procedure until we have a score for all voxels and all model layers. No hyperparameter selection was performed over the course of the benchmarking, apart from a minimal, exploratory grid search for a lambda parameter (of values: $1e1, 1e2, 1e3, 1e4, 1e5, 1e6, 1e7$) on an AlexNet model that we subsequently excluded from the main analysis. Thus, all feature spaces were projected to 5960 sparse random projections, and all regressions were run with a lambda penalty of $1e5$. The generalized (leave-one-out) cross-validation procedure in this case was used exclusively for evaluating model performance.

Rather than taking single-voxel fits as our key measure, we consider the geometry of the population across the larger region of interest as a critical target for prediction. To do so, we use the predicted responses from our voxel-wise encoding method to generate predicted representational dissimilarity matrices. The logic behind this procedure is effectively to dispense with or otherwise transform irrelevant features from the network via reweighting, such that new images are cast into a weighted subspace of the original feature space. The representational geometry of this subspace serves as the comparison to the brain. At the end of this procedure, we obtain a matrix of correlation scores of dimension *number of subjects x number of ROIs x number of model layers x number of models*.

## A.5 Intersubject Predictivity and the Noise Ceiling

In general, the purpose of a noise ceiling is to estimate (at the level of an individual unit of prediction) how reliable the response in that unit is across time. This metric allows us to then quantify how well our response data at one point predicts our response data at another. One example of such a measure relevant to fMRI is the Spearman-Brown-corrected split-half reliability of a voxel response over sequential presentations of the same stimuli. However, this method tends to underestimate true voxel reliability in regimes with few presentations.

The alternative we have provided here – the pairwise inter-subject representational similarity reference – is straightforward in its calculation, and computationally equivalent to the procedure for

---

[3]Note that after the SRP procedure there is no longer an interpretable mapping between individual model features and brain voxels; nonetheless, we have confirmed empirically that SRP procedure yields similar brain predictivity compared to a control analysis using AlexNet, a model whose feature map dimensionality is sufficiently low to run our encoding procedure without SRP.

benchmarking the models with classical RSA (which is to say, that subject RDMs and model RDMs were computed in the exact same way, and compared using the same correlation metric).

As a reference point for weighted RSA, however, this threshold is a bit misleading – since only the models benefit from the reweighting. One possible alternative, similar to work done recently in the neural network modeling of mouse visual cortex (Nayebi et al., 2021), is to directly incorporate the activity of other human brains into a regression procedure wherein the regressand is the brain activity of a target subject and the regressors are the brain activities of other subjects. This procedure has the advantage of equating each step in the analytic pipeline that maps model feature spaces to the brain, and of providing similarly intuitive targets that undergird inferences over how much of the variance in a target biological system we can capture with a system that is not biological.

As a preliminary test of this *reweighted* inter-subject predictivity, we consider another version of the pairwise metric above, predicting single subjects using data from other single subjects. For each pair of subjects, in which one is the target and the other is the contrast, we iterate over ROIs, gathering all voxels from the contrast's ROI to serve as regressors in the prediction of activity in each of the target's ROI voxels. We repeat this procedure until we have predicted all voxels in a given target subject with all possible contrast subjects. The mapping procedure in this case is exactly the same as it was for the mapping of models, controlled even to the hyperparemeter: we project the ROI voxel activity from the contrast subject to 5960 sparse random projections, and regress these subjects to the target voxel with a ridge regression set to a lambda penalty of $1e5$.

While this method equates each step in the mapping between model and human, the intersubject predictivity threshold it establishes is even lower than the version without reweighting. One reason for this may be that the brain activity from a single subject does not provide a sufficient breadth of variance to benefit from the reweighting. As a first pass at rectifying this issue, we devised a new measure, predicting each voxel from the *concatenated* activity of all voxels from all other subjects in the target ROI, effectively creating a multi-human reference. While we are continuing to assess, conceptually, whether such a reference point is useful, the estimates it produces for individual subjects are indeed far higher than the estimates of either the unweighted individual subject-to-subject comparison or the corresponding weighted comparison, and is in most cases far higher than the observed levels of model prediction. Figure 5 shows a comparison between the different kinds of human reference points we compute.

### A.6  Generalization Scores across Subject + ROI

Figure 4 shows the generalization scores across individual subjects and individual ROIs.

### A.7  Compute Required

We used a single machine with 8 Nvidia RTX 3090 GPUs, 755gb of RAM, and 96 CPUs. GPUs were used only for extracting model activations, and could (without major slowdown) be removed from the analytic pipeline. Dimensionality reduction and regression computations were CPU and RAM intensive. Replicating all of our results would take approximately two weeks on a similar machine.

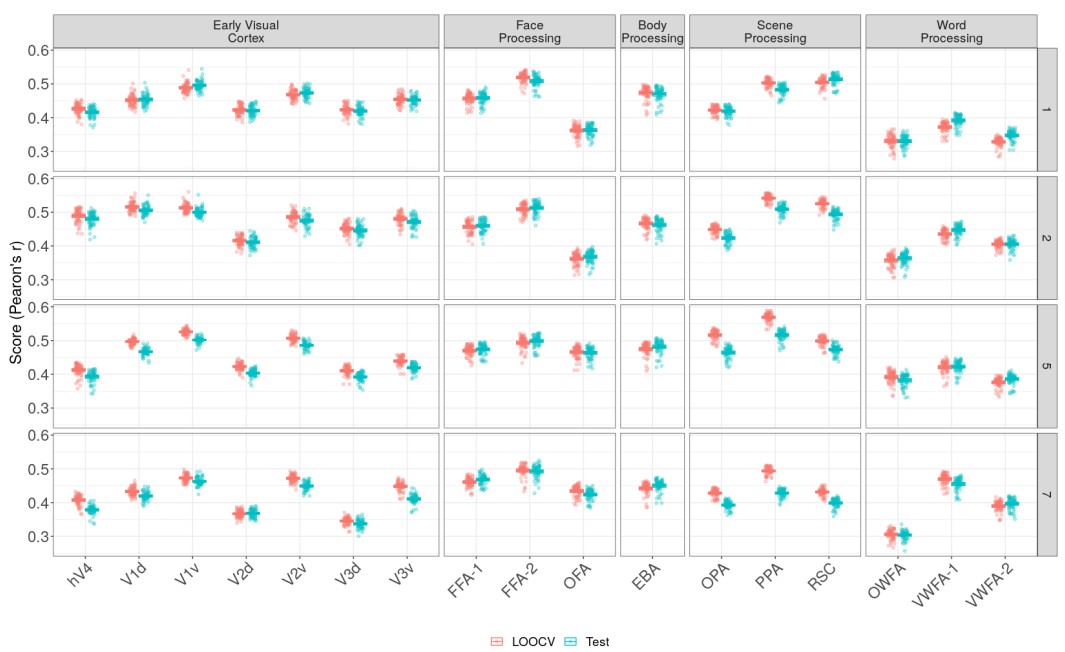

Figure 4: *Generalization scores across subject and ROI. Each point in red is the LOOCV score for a given model over the 1000 training images; each point in blue is the generalization to 1000 unseen images never incorporated into the training procedure.*

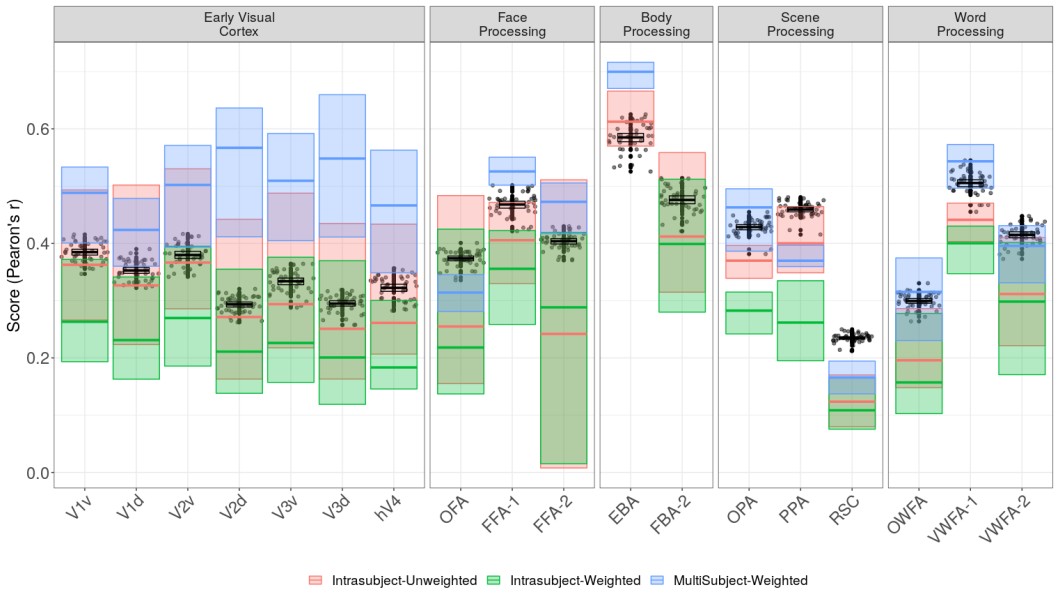

Figure 5: *A plot demonstrating different kinds of human reference points. Points in black are model scores from the voxelwise-encoding RSA method (a method that involves reweighting). The colored bands are the 95% confidence interval on a given reference metric. Reweighting the neural activity of multiple subjects to predict a single target subject (in blue) is the one reference point that reweighted models do not consistently outperform.*

