# OpenReview forum: "What can 5.17 billion regression fits tell us about artificial models of the human visual system?"
_NeurIPS.cc/2021/Workshop/SVRHM — SVRHM 2021 Poster_

### Official Review · Reviewer_qsNA · 2021-10-20
**A large-scale experiment that confirms several existing findings**

**Rating:** 6
**Confidence:** 4

**Review:**

## Summary
The authors conduct a large-scale experiment into the features of DNNs responsible for their success in explaining neural activity in visual cortex. Although similar studies have been conducted, this work primarily differs in that it considers a large human fMRI dataset with natural stimuli. They consider the effects of model architecture, training task, and feature reweighting on RSA performance.

Overall, I found this an interesting and well-written paper that was methodologically sound and provided convincing findings. My primary concern is in regards to novelty, as the authors' main conclusions have largely been demonstrated in one form or another elsewhere.

## Positives
- Due to the large-scale nature of the experiment and the use of well-established techniques in the field, the results are convincing. Methodologically, I have no problems with the work.
- Diverse existing findings in the literature, such as the effect of architecture and training task, are reproduced under a single analysis pipeline. The paper thus provides a useful lay-of-the-land for the current state of deep learning models of human vision.
- The methods and findings are clearly presented, as are the figures.
- There computational challenge of fitting so many models was significant, and therefore all the more valuable. Release of code and data will also be valuable to the community for further studying models of biological vision.

## Negatives
- In terms of approach and high-impact findings, the paper lacks novelty. It resembles BrainScore in many ways, and it is unclear why a meta-analysis of BrainScore-submitted models would have been insufficient or would have led to different conclusions. Granted, this study looks at human fMRI data rather than monkey electrophysiology, but it's not clearly argued if or why this would yield qualitatively different results.
- The effect of task-type (e.g. classification, contrastive-learning, autoencoding) has already been characterized, with the same conclusion that classifcation and contrastive-learning models outperform others.
- The irrelevance of architecture is overstated. The authors only consider 3 classes of architectures (ConvNets, Vision Transformers, and MLP-mixers), which is not sufficient to claim that architectural differences have little consequence. What about MLPs, RNNs that look at pixel sequences, or recurrent models in general? Moreover, although the 3 architectures considered appear to differ dramatically, they may in fact share inductive biases on some level (for instance, they all operate on patch representations, which has recently been shown to be [sufficient for good classification performance](https://openreview.net/forum?id=TVHS5Y4dNvM)).
- Downstream task performance is known to be deeply related to brain predictivity, but this is never mentioned nor looked at in the paper. To what degree does task performance (e.g. ImageNet transfer) explain the authors' results across models and brain regions.
- One of the most significant gaps in model performance is between the ImageNet-trained models and the Taskonomy-trained ones. This suggests that the training dataset is one of the primary factors responsible for brain similarity. However, the authors never state this observation, and it is only noticed when comparing two separate figures. Which factors of a dataset are relevant? Its size? Its diversity? Its naturalism? These questions are never discussed, even though the finding is present in the authors' own results.

---

### Official Review · Reviewer_47EP · 2021-10-26
**An informative quantitative survey, but there's still work ahead**

**Rating:** 7
**Confidence:** 4

**Review:**

This study evaluated the representational geometries of 72 DNN models of diverse architectures and tasks against the visual responses to 1000 COCO images in the NSD dataset. The authors used two measures of correspondence between model and brain representation: representational similarity analysis and representational similarity analysis applied to voxel-wise encoding models. Unsurprisingly, the latter approach results in higher RDM correlations. In terms of model comparison, CONVnets showed a modest advantage over vision transformers and MLP mixers, but only when the models were evaluated without data fitting (i.e., "classical RSA"). Task proved to be a more important factor: the task of object classification led to better model-brain fits, superseded only by self-supervised learning. This result generalized across the two RSA methods.

Overall, this work is an informative survey on how SOTA neural networks relate to human visual responses. My comments are below.

1. The research context is incomplete. Khaligh-Razavi, Henriksson, Kay, and Kriegeskorte (2017, J Math Psychol., https://dx.doi.org/10.1016%2Fj.jmp.2016.10.007) wrote an entire paper on the distinction between the two RSA measures the authors considered here and applied them to study DNNs and fMRI responses (although at a far smaller scale than the current study). Furthermore, the comparison between these two methods appears even earlier: consider Figure 7 in Cadieu, Hong and colleagues (2014, PLOS COMP BIO, https://doi.org/10.1371/journal.pcbi.1003963). I am not stating this to imply a lack of novelty, but the paper is not well placed in the literature.

2. Different models might have considerably different optimal ridge regression lambda values. The predetermined hyper-parameter choice might have biased the model comparison toward models of particular dimensionality.

3. The authors should consider testing whether the sparse regression approach yields the same model comparison hierarchy as alternatives such as the unreduced ridge regression or separable models (e.g., St-Yves & Naselaris, 2018, NeuroImage, https://doi.org/10.1016/j.neuroimage.2017.06.035).

4. As the authors note, the between-subjects RDM correlation is an inappropriate noise ceiling bound for models fitted and tested within a subject. As it is, the right-hand panels of figure 2 are misleading. A straightforward solution would be to evaluate the RDM predictions across subjects rather than within subjects.

5. Testing on completely held-out images is the (only) correct evaluation procedure. There's no justification for analyzing data affected by selection bias, even if this bias is small. In particular, not accounting for the fact that multiple layers were considered unfairly favors deeper models. Therefore, it would be more rigorous to report only the analyses conducted on fully held-out-data (or on data tested with nested cross-validation).

6. Considering the entire range of stimuli in NSD would make this work stronger and more extensive.

---

### Official Review · Reviewer_qitZ · 2021-10-28
**A well written paper with somewhat unclear implications**

**Rating:** 6
**Confidence:** 3

**Review:**

The authors of the paper benchmark 72 neural networks to understand if they can be used to improve our understanding of the human visual system. They find two key results. First, that neural net architecture does not well correlate with the predictive power of the network. Secondly, different visual tasks lead to different network performances.

Overall, the results are reasonably clear and the quality of the research is good. I have some concerns that there is little advancement in knowledge as a result of this paper, and the exact implications for future research are not clear. In general, the claims made in the abstract are not particularly clearly supported by the results. However, the amount of work and effort are noted and I can recommend it for acceptance to SVRHM.

Pros: Clearly written results, interesting insight in discussion, high quality figures

Cons: Lack of scientific advancement, unclear impact on future work

---

### Official Review · Reviewer_mueB · 2021-10-31
**Impressive deep net vs. visual system survey – methods need more clarification**

**Rating:** 6
**Confidence:** 2

**Review:**

This is a large-scale survey of dozens of deep neural network models on the recent Natural Scenes Dataset, and as such a suitable contribution to this workshop. 72 neural networks have been analysed for brain similarity for 16 visual system regions, using regular RSA and veRSA (building encoding models and subsequent RSA).

I find several results that can be seen in the figures quite interesting, particularly that object and scene recognition networks still have the best performance among all the ones surveyed. The amount of effort going into this survey should be appreciated. The manuscript should still be improved in clarity of methods and investigation of results.

I have a few points:

* It is unclear to me how hyperparameters for the veRSA regression models were selected. During the leave-one-out cross-validation? Or has the cross-validation been used to determine model performance? Or for both? (then it should be nested)

* Related: In encoding models using ridge regression the voxel-wise lambda parameter can have a notable influence on the outcome. In the supplementary materials it is stated (line 390) that no hyperparameter search was performed for the regression models, but that they were estimated once on AlexNet and then applied on all other networks. There should be an analysis on a subset of neural network models as to whether this has an influence. It is likely that higher performance could be obtained with a hyperparameter search. It is unclear whether this influences the observed differences between the models.

* I understand the veRSA method from (11) as been used, not FR-RSA from (12). As both methods are similar in spirit, could this be clarified in text?

* It is unclear to me whether the 1000 images from the held-out set were only used during section 3.4 "Generalization Test". Have they only been used as a test set for the the regression models? How were these 1000 images selected?

* For such a large survey, across many models and visual brain areas, there is too little specific discussion and close inspection of the results right now. The discussion section makes very general statements and could benefit from more specific observations.

* There is recent work surveying neural networks from the Taskonomy project, which has also been used in this study. Please consider citing [1] (presented at COSYNE 2021). Also perhaps consider embedding [2] in the literature overview, as a well-known early survey.

* The title seems inappropriate. While the effort that has gone into this survey is impressive, half of the results were obtained with regular RSA which does not involve regression fits. It would be better to use the title for stating the most important result instead, such as that object recognition networks are the best performing ones across the many tasks studied.

* The writing is frequently trading clarity for expressions that seem out-of-place (resounding affirmative, recent advents) and thus hinder understanding. Please prefer crystal clear and common, simple words to communicate what has been done.

[1] Cadena et al. (2021): "A diverse task-driven characterization of early and mid-level representations of the primate ventral stream".

[2] Khaligh-Razavi et al. (2014). Deep supervised, but not unsupervised, models may explain IT cortical representation.

---

### Decision · Program_Chairs · 2021-11-02

Accept (Poster)